# Modeling Bland–Altman Limits of Agreement with Fractional Polynomials—An Example with the Agatston Score for Coronary Calcification

Oke Gerke [1,2,*] and Sören Möller [2,3]

1 Department of Nuclear Medicine, Odense University Hospital, 5000 Odense, Denmark
2 Department of Clinical Research, University of Southern Denmark, 5000 Odense, Denmark; soren.moller@rsyd.dk
3 Open Patient Data Explorative Network, Odense University Hospital, 5000 Odense, Denmark
* Correspondence: oke.gerke@rsyd.dk

**Abstract:** Bland–Altman limits of agreement are very popular in method comparison studies on quantitative outcomes. However, a straightforward application of Bland–Altman analysis requires roughly normally distributed differences, a constant bias, and variance homogeneity across the measurement range. If one or more assumptions are violated, a variance-stabilizing transformation (e.g., natural logarithm, square root) may be sufficient before Bland–Altman analysis can be performed. Sometimes, fractional polynomial regression has been used when the choice of variance-stabilizing transformation was unclear and increasing variability in the differences was observed with increasing mean values. In this case, regressing the absolute differences on a function of the average and applying fractional polynomial regression to this end were previously proposed. This review revisits a previous inter-rater agreement analysis on the Agatston score for coronary calcification. We show the inappropriateness of a straightforward Bland–Altman analysis and briefly describe the nonparametric limits of agreement of the original investigation. We demonstrate the application of fractional polynomials, use the Stata packages *fp* and *fp_select*, and discuss the use of degree-2 (the default setting) and degree-3 fractional polynomials. Finally, we discuss conditions for evaluating the appropriateness of nonstandard limits of agreement.

**Keywords:** fractional polynomial regression; logarithm; method comparison; observer; rater; reliability; repeatability; reproducibility; square root; transformation

**MSC:** 92B15



## 1. Introduction

Bland–Altman limits of agreement, or simply the Bland–Altman plot, is a widely used technique in method comparison studies on continuous measurements. It consists of a simple scatterplot of paired differences against their respective averages, with an estimate for the bias (the estimated mean difference) and the so-called limits of agreement, which are equal to the estimated bias plus/minus 1.96 times the standard deviation of the paired differences [1]. The rationale behind this is the 68–95–99.7 rule of normally distributed variables: the limits of agreement are estimates for the range within which 95% of the population differences are expected to lie if the bias is constant and the differences are independent and identically normally distributed across the measurement range [2]. Whether the observed differences actually follow roughly a normal distribution can be checked visually by plotting a histogram of the differences and supplementing it with an approximate normal distribution using the empirical mean, m, and standard deviation, s. Moreover, one may assess the proportion of differences that fall within m ± s, m ± 2 s, and m ± 3 s and compare these with the expected proportions of 0.68, 0.95, and 0.997,

respectively, in light of the 68–95–99.7 rule for a normal distribution. Bland–Altman analysis is a simple technique to assess whether two methods measure sufficiently in agreement by comparing the limits of agreement with predefined clinically acceptable values. The seminal paper [3] has been cited about 40,129 times to date and is currently the most cited Lancet paper according to Scopus.

Bland and Altman [4] emphasized the precision of the estimated limits of agreement by providing approximate 95% confidence intervals. They also gave a more comprehensive and pedagogical account of using log transformation as a variance-stabilizing transformation before analysis, introducing a regression approach for non-uniform differences, measuring agreement using repeated measurements, and comparing methods with a nonparametric approach. Ludbrook [5] offered step-by-step guidance for Bland–Altman analysis and illustrated it visually with examples. Bland and Altman [6] examined method comparison with multiple observations per individual further, which Olofsen et al. [7] built on and implemented a freely accessible online assessment sheet [8]. Carkeet [9] proposed exact confidence intervals for Bland–Altman limits of agreement. Jan and Shieh [10] and Shieh [11,12] compared approximate 95% confidence intervals for the limits of agreement with exact ones and motivated, for example, sample size rationales on exact 95% confidence intervals that cover the central 95% proportion of the differences. Taffé [13–16] advised against Bland–Altman analysis if the precision of the two measurement methods differs or in the case of nonconstant bias. As an alternative graphical approach, he proposed a set of graphs that support the assessment of bias, precision, and agreement between two measurement methods. His method requires repeated measurements of at least one of the two methods to be compared with another. Gerke et al. [17] gave a brief overview of recent developments and recommendations on sample size considerations in method comparison studies.

Figure 1 shows a collection of four Bland–Altman analyses that were derived using different methods. The top left corner shows an example of a straightforward Bland–Altman inter-rater analysis with exact 95% confidence intervals for the limits of agreement, as proposed by Carkeet [9]. The top right corner is an example of a regression approach for non-uniform differences [4], the bottom left corner of deriving Bland–Altman limits of agreement on log-transformed measurements and transforming the limits of agreement back to the original scale [4]. Finally, the example in the bottom right corner shows limits of agreement that resulted from Bland–Altman analysis on square root-transformed measurements [18]. The latter limits of agreement were the outcome of a modeling process that started with fractional polynomial regression.

This review revisits a previous inter-rater agreement analysis of the Agatston score for coronary calcification [19] and explores an alternative assessment of the limits of agreement following the approach of Sevrukov et al. [18]. We conclude that obtaining nonstandard Bland–Altman limits of agreement may require a modeling process when straightforward Bland–Altman analysis is, given the scatter of differences against averages, inappropriate and an appropriate choice for a variance-stabilizing transformation is unclear. Finally, we discuss conditions for the appropriateness of nonstandard limits of agreement.

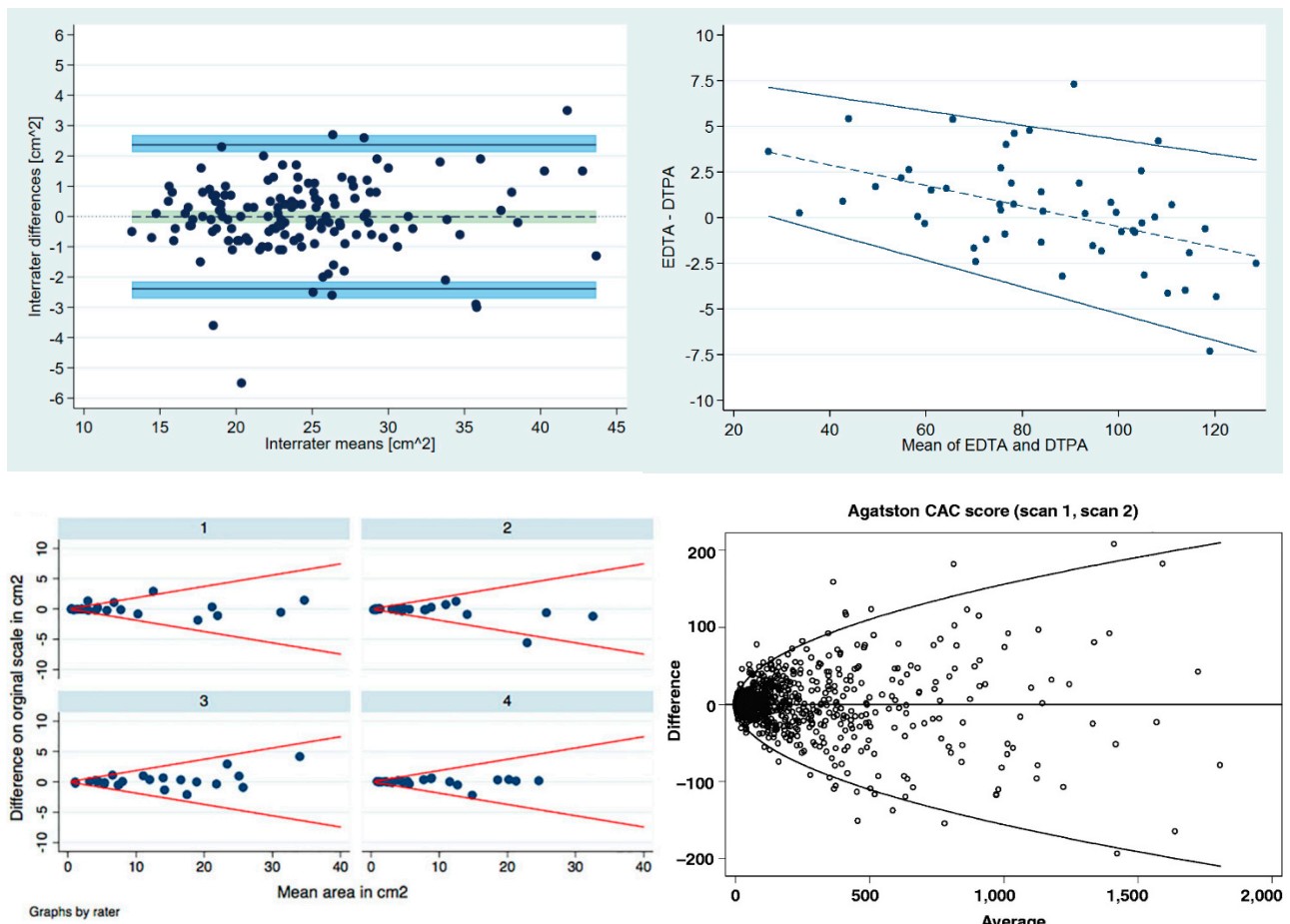

**Figure 1. Top left**: Bland–Altman plot for inter-rater agreement analysis of left atrium area (cm$^2$) measurements in non-contrast computed tomography, n = 140 (reproduced with permission from [20]). **Top right**: Bland–Altman plot of glomerular filtration rate measured with 51Cr-ethylenediamine tetraacetic acid (EDTA) and 99mTc-diethylenetriamine pentaacetic acid (DTPA), n = 51 (reproduced with permission from [21]). **Bottom left**: Intra-rater agreement assessment for 2D measurements (cm$^2$) for raters 1, 2, 3, and 4, respectively; n = 48 (reproduced with permission from [22]). **Bottom right**: Distribution of differences between repeated measurements of coronary artery calcium (CAC) as function of average CAC score expressed in Agatston CAC score units; the curve shows 95% repeatability limits which include 98% of differences, n = 2217 (reprinted from 'Serial electron beam CT measurements of coronary artery calcium: Has your patient's calcium score actually changed?', A.B. Sevrukov, J.M. Bland, and G.T. Kondos, the *American Journal of Roentgenology* 185, Copyright© 2023, copyright owner as specified in the *American Journal of Roentgenology* [18]).

## 2. Data

An intra-rater variation analysis of the CAC score was part of a previous study that aimed to calculate population-based CAC score percentiles in Danish women and men, 50–75 years of age [19]. These analyses were based on two Danish trials, namely, the Danish Cardiovascular Screening Trial (DANCAVAS) [23] and its precursor called the Danish Risk Score study (DanRisk) [24]. The CAC score has been shown to improve the discrimination and reclassification of coronary artery disease on top of the traditional risk factors, which are age, sex, smoking status, diabetes mellitus, blood pressure, hyperlipidemia, and race [25–29]. A low-dose computed tomography scan without contrast visualizes calcifications of any artery for which CAC scores are derived [23]. Since its original proposal back in 1990 [30], the CAC score has repeatedly been subject to agreement assessments in order to investigate the score's reliability. A recent review [31] found sample sizes to be highly variable in studies of agreement on the CAC score (10–9761), and research groups

focused on intra- and inter-rater as well as intra- and inter-scanner variability assessments. Andersen and Gerke [31] concluded that only very few research articles were capable of deriving limits of agreement that fit the observed data visually in a convincing way. This is why alternative methods like fractional polynomial modeling may prove useful in assessing the agreement of the CAC score.

Our dataset consists of 129 randomly selected participants of DanRisk and 101 randomly selected participants of DANCAVAS. The scatterplot of these 230 inter-rater differences against their respective means is shown in Figure 2. The distribution of differences is markedly over-represented with zeros (n = 101, 43.9%) and 12 out of 230 observations (5%) were associated with absolute differences exceeding 50 HU. Table 1 shows the descriptive statistics for the paired differences. The distribution was highly centered around 0, with flat tails and far from normal. The mean and the standard deviation of the 230 differences were −0.93 and 80.25, respectively.

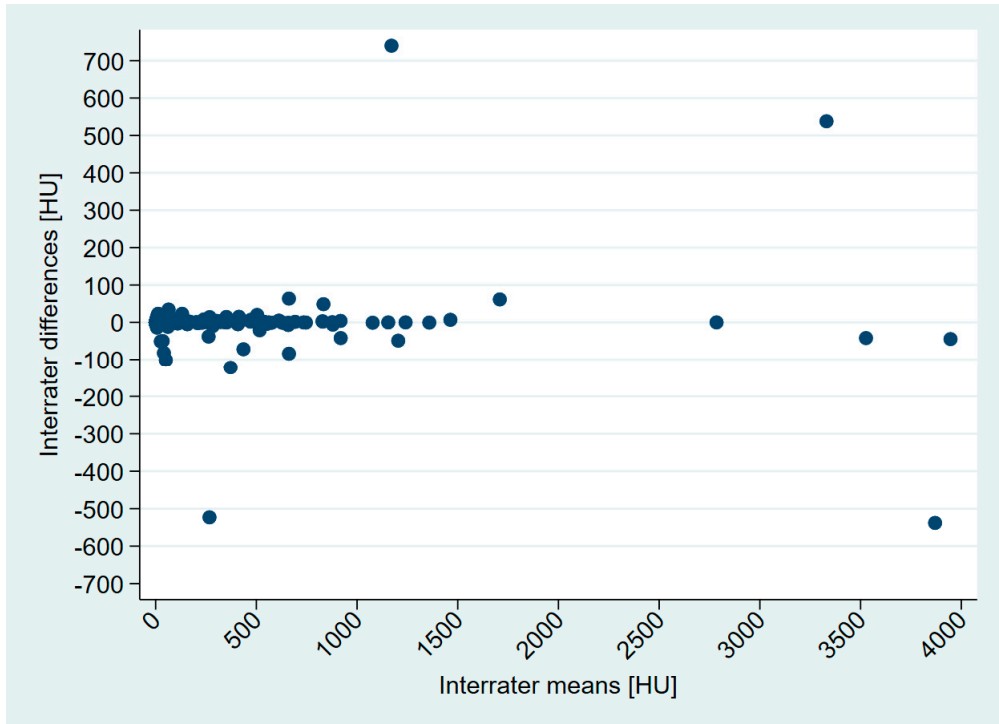

**Figure 2.** Mean difference plot for inter-rater variation analysis reported in [19].

**Table 1.** Descriptive statistics for the paired differences (n = 230).

| Minimum | P5 [1] | P25 | Median | P75 | P95 | Maximum |
|---|---|---|---|---|---|---|
| −538 | −42 | −0.3 | 0 | 0.4 | 15 | 740 |

[1] 5th quantile.

## 3. Bland–Altman Limits of Agreement and Previously Reported Nonparametric Limits of Agreement

Applying Bland–Altman analysis naively despite non-normally distributed differences results in Bland–Altman limits of agreement of −158.22 and 156.36 (Figure 3, left). Obviously, four paired differences that exceeded 500 HU in absolute terms widen the band that the Bland–Altman limits of agreement span. These four pairs of differences have a significant influence on the limits themselves.

As no obvious transformation to another scale of the measurements came to mind [32,33], nonparametric limits of agreement were derived for the inter-rater comparison in [19], which were −83 and 38 HU (Figure 3, right). These limits were estimated using a simple sample

quantile estimator for the 2.5% and 97.5% percentiles [34] and are—unlike Bland–Altman limits of agreement—by definition usually asymmetrical around the y-axis. This nonparametric estimator is a weighted average of the two paired differences that are closest to the target percentile. In other words: only four paired differences were used to estimate the 2.5% (lower limit of agreement) and the 97.5% percentile (upper limit of agreement). The estimated bias in terms of the median difference was 0 HU.

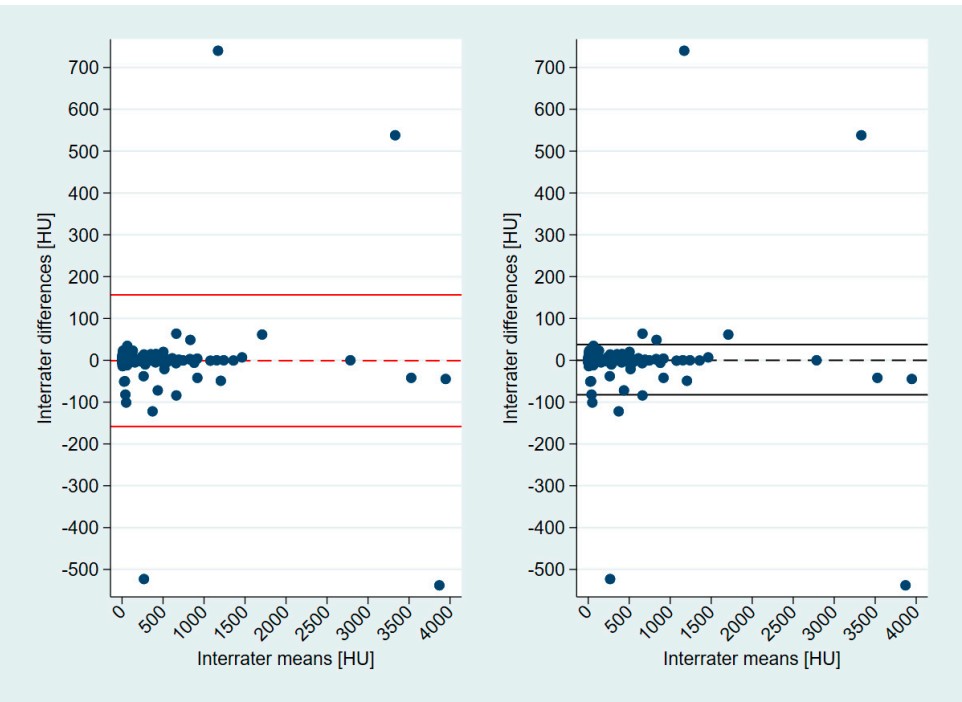

**Figure 3.** Bland–Altman limits of agreement (**left**) and nonparametric limits of agreement (**right**) reported in [19].

Bland–Altman limits of agreement (Figure 3, left) cover all paired differences except the four pairs mentioned above, that is, 226 out of 230 (98.26%). The asymmetrical nonparametric limits of agreement (Figure 3, right) cover 95.65% of the 230 paired differences.

## 4. Regression of Non-Uniform Differences on the Averages

Sevrukov et al. [18] observed that the size of the difference in their dataset increased with increasing average (see Figure 1, bottom right) and defined the repeatability of the measurement method as a function of the measurement size. They employed a regression approach for non-uniform differences [4,35], and the variation in the differences (D) between repeated measurements was modeled as a function of the measurement size, which in turn was estimated from the average (A) of the paired measurements. Sevrukov et al. [18] considered the absolute values of D, namely, |D|, and regressed |D| on A. The repeatability of the method only depends on the distribution of random measurement errors; therefore, D is normally distributed with mean zero for all A, and |D| is half-normally distributed with a mean that equals the standard deviation of |D| multiplied by $\sqrt{\pi/2}$ [35,36]. As a consequence, multiplying the mean value of |D| with $\sqrt{\pi/2}$ results in the standard deviation of the differences. Multiplying this standard deviation with the 97.5% percentile of the standard normal distribution, i.e., 1.96, leads to the 95% repeatability coefficient for any given value of A.

## 5. Fractional Polynomials

Royston and Altman [37] described a family of model functions for a single, positive covariate $X$ and defined a fractional polynomial of degree $m$ as

$$f_m(x, \boldsymbol{\beta}, \boldsymbol{p}) = \beta_0 + \beta_1 x^{(p_1)} + \beta_2 x^{(p_2)} + \ldots + \beta_m x^{(p_m)} \tag{1}$$

with a real-valued vector of powers $\boldsymbol{p} = (p_1, \ldots, p_m)$, and a real-valued vector of coefficients $\boldsymbol{\beta} = (\beta_0, \beta_1, \beta_2, \ldots, \beta_m)$. The round bracket notation indicates the Box–Tidwell transformation

$$x^{(p_j)} = \begin{cases} x^{(p_j)} & if \ p_j \neq 0 \\ \ln(x) & if \ p_j = 0 \end{cases} \tag{2}$$

.

The powers $p_j$ are chosen from a restricted set; S. Royston and Altman [37] suggested for pragmatic reasons $S = \{-2, -1, -0.5, 0, \ 0.5, \ 1, 2, 3\}$. Powers $p_j$ are allowed to repeat in fractional polynomials. Whenever a power repeats, it is multiplied by another $\ln(x)$. For instance, a fractional polynomial with $\boldsymbol{p} = (0, 0, 2)$ becomes

$$f_3(x, \boldsymbol{\beta}, \boldsymbol{p}) = \beta_0 + \beta_1 \ln(x) + \beta_2 \{\ln(x)\}^2 + \beta_3 x^2. \tag{3}$$

Figure 4 shows some examples of degree-1, degree-2, and degree-3 fractional polynomials for illustration purposes.

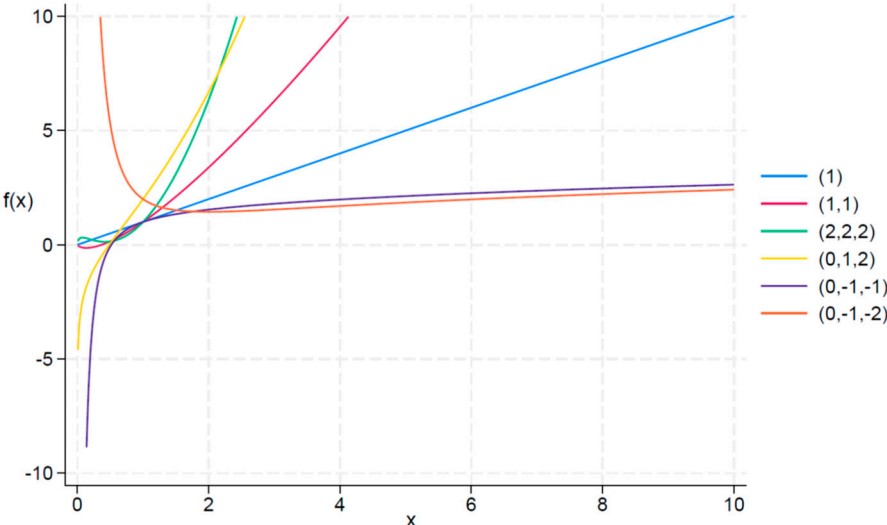

**Figure 4.** Examples of some functional forms available with degree-1, degree-2, and degree-3 fractional polynomials with various powers ($p_1$), ($p_1$,$p_2$), and ($p_1$,$p_2$,$p_3$), respectively.

Fractional polynomials increase the flexibility known from the family of conventional polynomials. Despite their popularity in data analysis, linear and quadratic functions are limited in their range of curve shapes. Cubic and higher-order curves may produce undesirable artifacts. Fractional polynomials are different from regular polynomials as they allow logarithms, non-integer powers, and powers to be repeated [38].

Sauerbrei et al. [39] revisited the approach in terms of software implementation in SAS, Stata, and R. In Stata, the package *fp* (fractional polynomial regression) fits degree-2 fractional polynomial models (that is $m = 2$) by default and chooses the fractional powers from the set $S = \{-2, -1, -0.5, 0, \ 0.5, \ 1, 2, 3\}$.

A search through all possible fractional polynomials up to degree-2 and with the powers set S is performed. Later, *fp* was supplemented by the post-estimation command *fp_select* [40]. Taking the results from the most recent run of *fp*, *fp_select* tries to simplify the most complex reported fractional polynomial model by applying an ordered sequence of

significance tests to reduce possible overfitting. We have used *fp* and *fp_select* on our data, as follows in Section 7.

**6. Analysis Strategy of Sevrukov, Bland, and Kondos**

Sevrukov et al. [18] started predicting the absolute difference $|D|$ of the repeated measurements of CAC using fractional polynomials and $S = \{0.5, 1\}$ :

$$|D| = -0.04632 + 1.488\sqrt{A} + 0.02393\,A. \tag{4}$$

Inspection of Equation (4) led Sevrukov et al. [18] to omit the final term because the main predicting variable was a square root transform of A. Regressing $|D|$ on A resulted in

$$|D| = -0.9733 + 2.067\sqrt{A}. \tag{5}$$

The residual mean square of model (5) was 186.96 and, hence, only slightly increased compared to the residual mean square of the polynomial regression of model (4) of 183.44. As the residual variance was almost identical, Sevrukov et al. [18] retained the simpler model (5). Beyond that, they forced the constant term in the regression model (5) to be zero in order to avoid negative standard deviations at small values of A. Model (5) simplified, then, to

$$|D| = 2.007\sqrt{A} \tag{6}$$

with a residual variance of 187.53. This slight increase in residual variance was deemed a marginal price to pay for model simplicity when moving from models (4)–(6). Therefore, they regressed the absolute differences on square root-transformed averages $(S = \{0.5\})$ and forced the intercept to be zero in their final model.

As described in the previous section, the standard deviation of the differences resulted then from multiplying the mean value of $|D|$ by $\sqrt{\pi/2}$:

$$SD_{|D|} = \left\{2.007\sqrt{A}\right\}\sqrt{\pi/2} = 2.515\sqrt{A}, \tag{7}$$

and the repeatability coefficient, *r*, from multiplying $SD_{|D|}$ by 1.96:

$$r = 1.96\left\{2.007\sqrt{A}\right\}\sqrt{\pi/2} = 4.930\sqrt{A}. \tag{8}$$

Figure 1 (bottom right) shows the resulting parabola-shaped repeatability limits based on model (6) and reported by Sevrukov et al. [18].

**7. Reanalysis of Inter-Rater Agreement Reported in [19]**

Revisiting the former inter-rater agreement analysis by Gerke et al. [19], we followed the analysis thread along the lines of Sevrukov et al. [18], but applied the readily available Stata packages *fp* and *fp_select* as a starting point. We have attached our Stata codes, the dataset, and our output as File S1, File S2, and File S3, respectively.

*7.1. Degree-2 Fractional Polynomial Models*

Figure 5 shows the Stata output for the default setting $m = 2$ that implies the use of degree-2 fractional polynomial models. The *fp* package investigated 44 models and proposed the set $S = \{0.5, 2\}$ for $m = 2$. The post-estimation package *fp_select* indicated that a simpler model, just including a linear term ($S = \{1\}$ for $m = 1$), may be equally sufficient while reducing the complexity of the model.

For $m = 2$ ($S = \{0.5, 2\}$), Equation (1) becomes here

$$|D| = 0.6248949 + 1.029373\sqrt{A} + 0.0000143\,A^2. \tag{9}$$

Consequently, the standard deviation of the differences resulted from multiplying the mean value of $|D|$ by $\sqrt{\pi/2}$:

$$SD_{|D|} = \left\{ 0.6248949 + 1.029373\sqrt{A} + 0.0000143\,A^2 \right\}\sqrt{\pi/2}, \tag{10}$$

and the repeatability coefficient, $r$, from multiplying $SD_{|D|}$ by 1.96:

$$r = 1.96\left\{ 0.6248949 + 1.029373\sqrt{A} + 0.0000143\,A^2 \right\}\sqrt{\pi/2}. \tag{11}$$

Figure 6 (top right) shows the resulting repeatability limits based on model (9). The coverage of these limits of agreement was 91.30%, i.e., considerably below 95%.

```
. fp <mean1>, dimension(2): regress absdiff <mean1>
(fitting 44 models)
(....10%....20%....30%....40%....50%....60%....70%....80%....90%....100%)

Fractional polynomial comparisons:
```

| mean1 | Test df | Deviance | Residual std. dev. | Deviance diff. | P | Powers |
|---|---|---|---|---|---|---|
| omitted | 4 | 2659.026 | 78.550 | 57.121 | 0.000 | |
| linear | 3 | 2603.657 | 69.794 | 1.751 | 0.633 | 1 |
| m = 1 | 2 | 2603.657 | 69.794 | 1.751 | 0.425 | 1 |
| m = 2 | 0 | 2601.905 | 69.682 | 0.000 | -- | .5 2 |

```
Note: Test df is degrees of freedom, and P = P > F is sig. level for tests
      comparing models vs. model with m = 2 based on deviance difference,
      F(df, 225).
```

| Source | SS | df | MS | | |
|---|---|---|---|---|---|
| Model | 310726.502 | 2 | 155363.251 | Number of obs = | 230 |
| Residual | 1102217.63 | 227 | 4855.58426 | F(2, 227) = | 32.00 |
| | | | | Prob > F = | 0.0000 |
| | | | | R-squared = | 0.2199 |
| | | | | Adj R-squared = | 0.2130 |
| Total | 1412944.13 | 229 | 6170.0617 | Root MSE = | 69.682 |

| absdiff | Coefficient | Std. err. | t | P>\|t\| | [95% conf. interval] | |
|---|---|---|---|---|---|---|
| mean1_1 | 1.029373 | .5150383 | 2.00 | 0.047 | .0145052 | 2.04424 |
| mean1_2 | .0000143 | 3.42e-06 | 4.19 | 0.000 | 7.59e-06 | .0000211 |
| _cons | .6248949 | 6.260334 | 0.10 | 0.921 | -11.7109 | 12.96069 |

```
. fp_select, alpha(0.05)

selected FP model: powers = (1), df = 3
```

**Figure 5.** Stata output for $m = 2$.

For the simpler model $m = 1$ with $S = \{1\}$, Equation (1) turns into

$$|D| = 0.6730657 + 0.0628848A. \tag{12}$$

The resulting limits of agreement cover only 84.78% of the observed differences (Figure 6, top left), which falls unacceptably short of 95%.

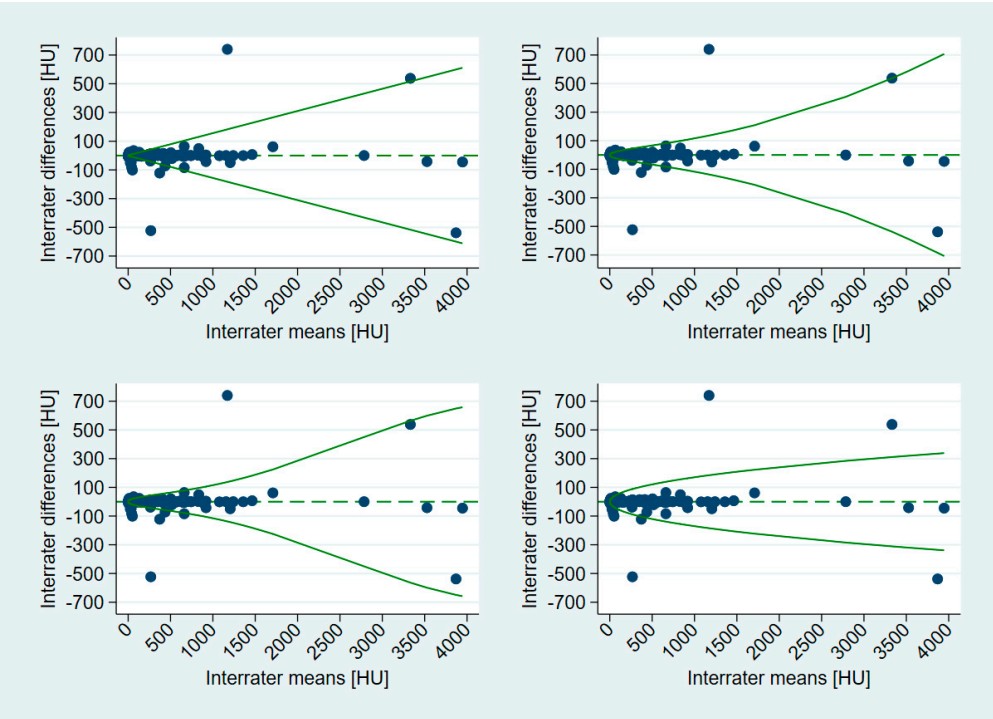

**Figure 6.** Limits of agreement based on fractional polynomial regression models. **Top left**: m = 1, $S = \{1\}$, coverage: 84.78%. **Top right**: $m = 2$, $S = \{0.5, 2\}$, coverage: 91.30%. **Bottom left**: $m = 3, S = \{0.5, 3, 3\}$, coverage: 91.74%. **Bottom right**: $m = 1, S = \{0.5\}$, coverage: 95.22%.

### 7.2. Degree-3 Fractional Polynomial Models

Increasing the allowed complexity by moving from degree-2 fractional polynomial models (*m* = 2; default setting) to degree-3 fractional polynomial models (*m* = 3) makes Stata evaluate 164 possible models (Figure S1). The proposed degree-3 fractional polynomial model was $S = \{0.5, 3, 3\}$, the coverage was 91.74%, and the resulting limits of agreement are shown in Figure 6, bottom left. Again, the post-estimation package *fp_select* indicated that a simpler model, just including a linear term ($S = \{1\}$ for *m* = 1), may be equally sufficient while reducing the complexity of the model.

### 7.3. Sevrukov, Bland, and Kondos Model

For the sake of comparison with the final model of Sevrukov et al. [18], we evaluated the analogous model regressing the absolute differences on square root-transformed averages and forced the intercept to be zero:

$$|D| = 2.193837\sqrt{A} \tag{13}$$

The respective parabola-like limits of agreement are shown in Figure 6, bottom right. These limits cover 95.22% of the paired differences.

## 8. Discussion

### 8.1. Main Findings

Sevrukov et al. [18] quantified the coverage of paired differences by their limits of agreement very conservatively as 98% (Figure 1, bottom right). In straightforward Bland–Altman analysis, where the assumptions of normality, constant bias, and variance homogeneity hold, the coverage is by definition roughly 95% (see, for instance, top left of Figure 1 with a coverage of 93.6%).

For the limits of agreement that were based on fractional polynomial regression models on our data, the coverage varied hugely: 84.78% for $S = \{1\}$, 91.30% for $S = \{0.5, 2\}$, 91.74% for $S = \{0.5, 3, 3\}$, and 95.22% for $S = \{0.5\}$ (Figure 6). The former three models employ

a constant term as per the definition of fractional polynomial models, whereas the latter model by Sevrukov, Bland, and Kondos [18] forces the constant to be zero. We increased the flexibility and the complexity by moving from the default setting of *fp* in Stata of degree-2 to degree-3 fractional polynomial models and identified $S = \{0.5, 3, 3\}$ (Figure 6, bottom left) as the final model. However, its coverage of 91.74% was actually considerably lower than both the nominal target level of 95% and the coverage of the Sevrukov-like model based on a square root transformation (95.22%, Figure 6, bottom right).

### 8.2. How Good Is Good Enough?

Appropriate *classical* Bland–Altman limits of agreement fit the observed point cloud, the bias is constant, the variance in the differences is homogeneous across the measurement range, and the sampled data cover the measurement range of interest sufficiently [20]. The limits of agreement cover (roughly) 95% of paired differences by definition due to the 68–95–99.7 rule [2]. For *nonstandard* limits of agreement, we propose the following two conditions to be useful for practical purposes:

1. The coverage of the observed differences should be roughly around 95%, in line with *classical* Bland–Altman limits of agreement.
2. The limits of agreement fit the data nicely and harmonize with the point cloud of paired differences.

For our data, the simple model with $S = \{0.5\}$ (see Figure 6, bottom right) based on a square root transformation (as in Sevrukov et al. [18]) fulfills the first and to some extent the second condition. The much more complex degree-3 fractional polynomial model with $S = \{0.5, 3, 3\}$ falls short of the first condition but fulfills the second condition and covers even the extreme observations (3331, 538) and (3870, −538); however, you may likewise argue that this is simply the result of an overfitting model. Bland–Altman limits of agreement (Figure 3, left) fulfill the first, but not the second condition. The originally proposed nonparametric limits of agreement (Figure 3, right) fulfill both conditions.

### 8.3. Strengths and Limitations

In Sevrukov et al. [18], the relationship between the difference and the average was nonlinear, and the size of the differences increased with increasing average of the paired measurements across the measurement range. This was not the case for our data, for which the distribution of differences was centered heavily around 0 and had a very flat tail. Still, deriving nonstandard limits of agreement with fractional polynomial regression offered visually appealing solutions with $S = \{0.5, 2\}$ and $S = \{0.5, 3, 3\}$ which, therefore, provides at least partly satisfying solutions with respect to conditions 1 and 2 above. For nonstandard limits of agreement, fractional polynomial modeling offers an alternative route of derivation, but the risk of overfitting simply irregularly scattered outliers (Figure 6, bottom left) exists and warrants caution in practice.

### 8.4. Future Perspectives

Previous pedagogical papers [4,5,7] have advised on Bland–Altman limits of agreement that come in various shapes and forms. A tutorial on how to tackle nonstandard situations in which the assumptions for classical Bland–Altman analysis fail and cannot be fulfilled by appropriate transformation of the data is lacking. Such a guidance paper will provide examples with unusually distributed paired differences like our worked example on the Agatston score for coronary calcification. Despite the challenge of covering sufficiently many different distributions of paired differences, such a tutorial will provide an overview of what methods have been applied across different research fields in the literature.

## 9. Concluding Remarks

Straightforward Bland–Altman analysis requires roughly normally distributed differences, a constant bias, and variance homogeneity across the measurement range. In

cases that violate these assumptions, the application of the simple and readily interpretable classical Bland–Altman limits of agreement would be misleading and would misrepresent the data at hand. This was visualized for our data in Figure 3 (left), with heavily inflated limits that were affected by four outliers.

Variance-stabilizing transformations (natural logarithm, square root [4,5,32,33]) may enable Bland–Altman analysis on another scale, implying back-transformation to the original scale after analysis (e.g., Figure 1, bottom left). The half-normal method [4,35] is a powerful and simple method for estimating nonstandard limits of agreement in light of nonconstant bias and/or variance heterogeneity [18]. Sevrukov et al. [18] exemplified a modeling process for deriving nonstandard limits of agreement for repeated CAC data. We have reanalyzed a formerly reported inter-rater agreement evaluation [19] on CAC data that followed a quite different distribution than those of Sevrukov et al. [18].

Fractional polynomials offer considerable flexibility whenever variance-stabilizing transformations are difficult to find. As holds true for any automated solution, the proposed choice of powers by *fp* and *fp_select* requires thoughtful judgement by the user. For our data, the empirical coverage satisfied the nominal aim of 95% for square root-transformed data. We have proposed two intuitively appealing conditions for judging the appropriateness of nonstandard limits of agreement that will hopefully prove useful to this end.

**Supplementary Materials:** The following supporting information can be downloaded at: https://www.mdpi.com/article/10.3390/axioms12090884/s1, File S1: Stata source code ("analyse_2023_06_22.do"), File S2: worked example data in Stata format ("interrater.dta"), File S3: Stata output ("Stata output 22 June 2023.pdf"), Figure S1: Stata output for *m* = 3.

**Author Contributions:** Conceptualization, O.G.; data curation, O.G.; formal analysis, O.G.; investigation, O.G. and S.M.; methodology, O.G. and S.M.; project administration, O.G.; resources, O.G.; software, O.G.; supervision, S.M.; validation, O.G.; visualization, O.G.; writing—original draft preparation, O.G.; writing—review and editing, O.G. and S.M. All authors have read and agreed to the published version of the manuscript.

**Funding:** This research received no external funding.

**Data Availability Statement:** File S2 comprises the worked example data.

**Acknowledgments:** The authors would like to express their gratitude to Axel Diederichsen (Odense University Hospital, Denmark) for the permission to reuse the example data. The authors would like to thank three anonymous reviewers for their helpful comments on earlier versions of the manuscript. Finally, the authors thank Bing, an AI language model developed by Microsoft, for grammar and spelling checking.

**Conflicts of Interest:** The authors declare no conflict of interest.

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
