# Peer review of "Modeling Bland–Altman Limits of Agreement with Fractional Polynomials—An Example with the Agatston Score for Coronary Calcification"

_axioms, doi:10.3390/axioms12090884_

Round 1

Reviewer 1 Report

The manuscript is excellently composed and exhibits almost all the qualities of a standard scientific paper. The manuscript's structure is apt, the problem statement is well-executed, the limitations of existing models are thoroughly discussed, the approach taken to address the matter is convincing, and the study's results are significant. Overall, I don't have any practical critique for the manuscript. However, to further enhance its quality, I can suggest the following points:

1- I suggest presenting the information from section 2 (Data) as a table for better readability and organization.

2- The authors may provide the numerical order of the equations in lines 159, 163, and 168.

3- Could you confirm if the sentences in lines 169 to 173 are all sourced from reference [31]? If not, please provide an appropriate reference for this paragraph's second and third sentences.

4- To further examine the efficiency of the proposed model, could you please provide a real-world example?

5- There are no directions given for future research. After listing the limitations of the proposed approach, it would be beneficial also to offer suggestions for future directions to the readers.

Despite some grammatical errors, I found the manuscript to be quite intriguing. Given its valuable content, I think it would greatly benefit from being edited by an English language expert to enhance its effectiveness.

Author Response

1- I suggest presenting the information from section 2 (Data) as a table for better readability and organization.

RESPONSE: Done. We have added Table 1.

2- The authors may provide the numerical order of the equations in lines 159, 163, and 168.

RESPONSE: done.

3- Could you confirm if the sentences in lines 169 to 173 are all sourced from reference [31]? If not, please provide an appropriate reference for this paragraph's second and third sentences.

RESPONSE: Confirmed. We have deleted the first appearance of reference [31] in that paragraph for more clarity.

4- To further examine the efficiency of the proposed model, could you please provide a real-world example?

RESPONSE: The worked example of Section 7 is a real-world example from trial data which are attached as Supplemental Material 2.

5- There are no directions given for future research. After listing the limitations of the proposed approach, it would be beneficial also to offer suggestions for future directions to the readers.

RESPONSE: Thank you. We have reformulated Section 8 as discussion and added both a section on the limitations and on future directions.

Comments on the Quality of English Language

Despite some grammatical errors, I found the manuscript to be quite intriguing. Given its valuable content, I think it would greatly benefit from being edited by an English language expert to enhance its effectiveness.

RESPONSE: We have rechecked spelling and grammar with Microsoft’s Bing.

Reviewer 2 Report

In the manuscript, Dr. Oke Gerke and Dr. Sören Möller presented a comprehensive application example and detailed instruction for methods implementation. However, this work is out of the scope of the journal. Please consider other journals for publication.

In addition, please consider my comments and suggestions below before submitting to other journals.

1. The "necessary and sufficient" condition is more like a practical guidance. It's useful for implementing the methods. However, it's not an axiomatic theory. Please consider re-drawing your conclusion with cautious about the wording used.

2. The manuscript focused on CAC data analysis. However, the Bland-Altman analysis methods could differ when using different type of data. The authors may add "CAC" data analysis in the title, since there is no sufficient evidence presented in the paper demonstrating that any conclusion drawn could be able to generalize to a broader area. 

3. Figure 1 is split in two pages. There is no descriptions for 4 sub-figures at bottom left.

4. In 107, is there any rationale using 50 HU as a cutoff?

5. There is no need to display Stata outputs in main text, since it's not a paper developed new method and package/software. Please consider to have them in the supplementary materials.

6. Even though, the authors are focusing on CAC data analysis, a comprehensive discussion on other types of data could be welcomed.

7. Please reconsider the conclusion drawn from the work and rewrite the section 8 & 9.

Author Response

In the manuscript, Dr. Oke Gerke and Dr. Sören Möller presented a comprehensive application example and detailed instruction for methods implementation. However, this work is out of the scope of the journal. Please consider other journals for publication.

RESPONSE: We agree that our submission at AXIOMS my seem surprising at first glance; however, we have submitted our paper to AXIOMS’ Special Issue “Probability, Statistics and Estimation” (https://www.mdpi.com/journal/axioms/special_issues/3K07L6CWUF) for which we deem our work a good fit.

In addition, please consider my comments and suggestions below before submitting to other journals.

  1. The "necessary and sufficient" condition is more like a practical guidance. It's useful for implementing the methods. However, it's not an axiomatic theory. Please consider re-drawing your conclusion with cautious about the wording used.

RESPONSE: We acknowledge that the formulation of a necessary and sufficient condition was far stretched. We have reformulated the conditions, refrained from the necessary & sufficient condition terminology, and adjusted the conclusions appropriately.

  1. The manuscript focused on CAC data analysis. However, the Bland-Altman analysis methods could differ when using different type of data. The authors may add "CAC" data analysis in the title, since there is no sufficient evidence presented in the paper demonstrating that any conclusion drawn could be able to generalize to a broader area. 

RESPONSE: We agree that Bland-Altman analysis varies depending on the data at hand as visualized by the Bland-Altman plot. We have highlighted in the title that our worked example represents Agatston score data. The point of our paper is, though, that modelling with fractional polynomials may be another route to pursue when regularly used methods (as shown in Figure 1) fail.

  1. Figure 1 is split in two pages. There is no descriptions for 4 sub-figures at bottom left.

RESPONSE: We have moved a paragraph of the introduction to keep Figure 1 completely on page 3.

  1. In 107, is there any rationale using 50 HU as a cutoff?

RESPONSE: 50 HU were chosen to underline that only 5% of the differences were 50 HU or larger. One possible categorization of the Agatston score is no calcification (0 HU), mild calcification (1-99 HU), moderate calcification (100-399 HU), and severe calcification (400 HU or more). To this end, 50 HU is halfway the mild calcification category.

  1. There is no need to display Stata outputs in main text, since it's not a paper developed new method and package/software. Please consider to have them in the supplementary materials.

RESPONSE: We kept Figure 4 for pedagogical reasons, but moved Figure 6 to the Supplemental Materials (S4).

  1. Even though, the authors are focusing on CAC data analysis, a comprehensive discussion on other types of data could be welcomed.

RESPONSE: It is difficult – if not impossible – to provide an exhaustive sketch of possible distributionals for the differences and the consequences on the consecutive Bland-Altman analysis thereof. With Figure 1, we intended to give some other, more often seen types in the literature, whereas modelling Bland-Altman Limits of Agreement with fractional polynomial modelling happens to be rarely applied so far. This review intends to raise more awareness of the possibilities that fractional polynomial modelling has to offer in agreement analysis. We have incorporated your input into our perspectives paragraph in the discussion.

  1. Please reconsider the conclusion drawn from the work and rewrite the section 8 & 9.

RESPONSE: We have reformulated Section 8 as discussion and added both a section on the limitations and on future directions. We have adjusted our conclusions by highlighting the risk of overfitting as a result of the modelling process in the limitations paragraph.

Reviewer 3 Report

Thank you for your submission. I appreciate the time and effort you have put into researching this important clinical issue.

Strengths:

1. The manuscript clearly explains the concept of Bland-Altman analysis and the assumptions required for the straightforward application of the method.

2. The example using coronary artery calcium scoring data helps demonstrate the limitations of naive Bland-Altman analysis on this non-normal data. 

Suggestions for improvement:

1. The introduction could provide more context about the clinical relevance of the worked example and the importance of inter-rater agreement studies for coronary calcium scoring. This would better motivate the methodological focus.

2. When introducing the fractional polynomial method, a bit more explanation or a figure showing how the curve shape varies with different power combinations could make this approach more accessible. The concept may be unfamiliar to some readers.

In summary, the manuscript gives a clear, worked demonstration of using fractional polynomials to derive nonstandard Bland-Altman limits of agreement when assumptions are violated. Providing more clinical context and intuitively explaining fractional polynomials could further strengthen the paper. This is a nicely explained methodological study of an important analytical technique.

Author Response

Thank you for your submission. I appreciate the time and effort you have put into researching this important clinical issue.

Strengths:

  1. The manuscript clearly explains the concept of Bland-Altman analysis and the assumptions required for the straightforward application of the method.
  2. The example using coronary artery calcium scoring data helps demonstrate the limitations of naive Bland-Altman analysis on this non-normal data. 

Suggestions for improvement:

  1. The introduction could provide more context about the clinical relevance of the worked example and the importance of inter-rater agreement studies for coronary calcium scoring. This would better motivate the methodological focus.

RESPONSE: Thank you for your encouraging and positive feedback. We have added a paragraph in lines 110-122 to this end.

  1. When introducing the fractional polynomial method, a bit more explanation or a figure showing how the curve shape varies with different power combinations could make this approach more accessible. The concept may be unfamiliar to some readers.

RESPONSE: Thank you. We agree and have added a new Figure 4 for illustration purposes.

In summary, the manuscript gives a clear, worked demonstration of using fractional polynomials to derive nonstandard Bland-Altman limits of agreement when assumptions are violated. Providing more clinical context and intuitively explaining fractional polynomials could further strengthen the paper. This is a nicely explained methodological study of an important analytical technique.

RESPONSE: Thank you! Much appreciated.

Round 2

Reviewer 2 Report

The authors addressed majority of my concerns. Please see my few questions below.

Comment #1

The authors reformulated the conditions, refrained from the necessary & sufficient condition terminology, and adjusted the conclusions appropriately. However, the "68-95-99.7 rule" mentioned is used to approximate the empirical data derived from a normal population. And the reason why the classical Bland-Altman analysis methods not working is that the data is not normally distributed. The authors could be more cautious to justify how the "68-95-99.7" rule could be used when classical Bland-Altman analysis methods not working.

Comment #3

The authors have updated the Figure 1. However, it's still not clear what 4 figures at bottom left refer to. Please consider to explain a little bit about sub-figure labeled as 1-4 at bottom left.

Author Response

The authors addressed majority of my concerns. Please see my few questions below.

Comment #1

The authors reformulated the conditions, refrained from the necessary & sufficient condition terminology, and adjusted the conclusions appropriately. However, the "68-95-99.7 rule" mentioned is used to approximate the empirical data derived from a normal population. And the reason why the classical Bland-Altman analysis methods not working is that the data is not normally distributed. The authors could be more cautious to justify how the "68-95-99.7" rule could be used when classical Bland-Altman analysis methods not working.

RESPONSE: Thank you for your encouraging and positive feedback. We have added lines 40-45 on normality assessment and use of the 68-95-99.7 rule to this end, thanks.

Comment #3

The authors have updated the Figure 1. However, it's still not clear what 4 figures at bottom left refer to. Please consider to explain a little bit about sub-figure labeled as 1-4 at bottom left.

RESPONSE: The subfigures 1-4 at bottom left of Figure 1 correspond to intra-rater analyses for rater 1, 2, 3, and 4 respectively. We have revised the legend accordingly to emphasize this point. Thank you.

Round 3

Reviewer 2 Report

The authors added description about normality assessment and the use of the 68-95-99.7 rule. However, the authors didn't provide the rationale how this rule could be used when the classical Bland-Altman analysis not working and data not normally distributed. It's fine, since it is not an article focusing on axioms, instead presenting application examples and implementation guidance.